# Molecular surveillance over 14 years confirms reduction of *Plasmodium vivax* and *falciparum* transmission after implementation of Artemisinin-based combination therapy in Papua, Indonesia

Zuleima Pava[1], Agatha M. Puspitasari[2], Angela Rumaseb[1], Irene Handayuni[1], Leily Trianty[2], Retno A. S. Utami[2], Yusrifar K. Tirta[2], Faustina Burdam[3,4], Enny Kenangalem[3,4], Grennady Wirjanata[1], Steven Kho[1], Hidayat Trimarsanto[2], Nicholas M. Anstey[1], Jeanne Rini Poespoprodjo[3,4,5], Rintis Noviyanti[2], Ric N. Price[1,6,7], Jutta Marfurt[1], Sarah Auburn[1,6,7]*

1 Global and Tropical Health Division, Menzies School of Health Research, Charles Darwin University, Darwin, Australia, 2 Eijkman Institute for Molecular Biology, Jakarta, Indonesia, 3 Mimika District Health Authority, Timika, Papua, Indonesia, 4 Timika Malaria Research Programme, Papuan Health and Community Development Foundation, Timika, Papua, Indonesia, 5 Pediatric Research Office, Department of Child Health, Faculty of Medicine, Public Health and Nursing, Universitas Gadjah Mada, Yogyakarta, Indonesia, 6 Centre for Tropical Medicine and Global Health, Nuffield Department of Clinical Medicine, University of Oxford, United Kingdom, 7 Mahidol-Oxford Tropical Medicine Research Unit (MORU), Faculty of Tropical Medicine, Mahidol University, Bangkok, Thailand

* sarah.auburn@menzies.edu.au

## Abstract

Genetic epidemiology can provide important insights into parasite transmission that can inform public health interventions. The current study compared long-term changes in the genetic diversity and structure of co-endemic *Plasmodium falciparum* and *P. vivax* populations. The study was conducted in Papua Indonesia, where high-grade chloroquine resistance in *P. falciparum* and *P. vivax* led to a universal policy of Artemisinin-based Combination Therapy (ACT) in 2006. Microsatellite typing and population genetic analyses were undertaken on available isolates collected between 2004 and 2017 from patients with uncomplicated malaria (n = 666 *P. falciparum* and n = 615 *P. vivax*). The proportion of polyclonal *P. falciparum* infections fell from 28% (38/135) before policy change (2004–2006) to 18% (22/125) at the end of the study (2015–2017); p<0.001. Over the same period, polyclonal *P. vivax* infections fell from 67% (80/119) to 35% (33/93); *p*<0.001. *P. falciparum* strains persisted for up to 9 years compared to 3 months for *P. vivax*, reflecting higher rates of outbreeding in the latter. Sub-structure was observed in the *P. falciparum* population, but not in *P. vivax*, confirming different patterns of outbreeding. The *P. falciparum* population exhibited 4 subpopulations that changed in frequency over time. Notably, a sharp rise was observed in the frequency of a minor subpopulation (K2) in the late post-ACT period, accounting for 100% of infections in late 2016–2017. The results confirm epidemiological evidence of reduced *P. falciparum* and *P. vivax* transmission over time. The smaller change in *P. vivax* population structure is consistent with greater outbreeding associated with

**Data Availability Statement:** All relevant data are within the manuscript and its Supporting Information files.

**Funding:** The study was funded by the Wellcome Trust (Senior Fellowship in Clinical Science to RNP, 200909 and ICRG GR071614MA) and the National Health and Medical Research Council of Australia (Improving Health Outcomes in the Tropical North: A Multidisciplinary Collaboration "Hot North" Career Development Fellowship to SA, grant number 1131932; Senior Principal Research Fellowship to NA, 1135820; and ICRG 283321); and supported by the Australian Centre for Research Excellence on Malaria Elimination (ACREME), funded by the National Health and Medical Research Council of Australia (1134989). SA is also supported by the Bill and Melinda Gates Foundation (OPP1054404) and a Georgina Sweet Award for Women in Quantitative Biomedical Science. The funders had no role in study design, data collection and analysis, decision to publish, or preparation of the manuscript.

**Competing interests:** The authors have declared that no competing interests exist.

relapsing infections and highlights the need for radical cure to reduce recurrent infections. The study emphasizes the challenge in disrupting *P. vivax* transmission and demonstrates the potential of molecular data to inform on the impact of public health interventions.

## Author summary

Genetic epidemiology is gaining widespread interest as a tool that can enhance conventional malaria surveillance. However, few studies have assessed the utility of molecular analyses in quantifying long-term changes in malaria transmission. The current study compared changes in the genetic diversity and structure of co-endemic *P. vivax* and *P. falciparum* populations sampled over 14 years (2004–2017) in Papua Indonesia, during which the incidence of both *P. falciparum* and *P. vivax* malaria halved. The study found larger genetic changes in *P. falciparum* than *P. vivax*, reflecting a greater impact of local interventions, including the implementation of a new drug policy (universal Artemisinin-Based Combined Therapy) in 2006, on *P. falciparum*. Both species exhibited decreasing complexity of infections over time, consistent with declining transmission. However, the *P. falciparum* population showed greater evidence of a recent bottleneck than the *P. vivax* population. Four subpopulations were observed amongst the *P. falciparum* isolates, one of which predominated in 2016–2017, potentially reflecting recent adaptation. The results concur with epidemiological studies performed in the same area, that found declining transmission in both species, with less impact on *P. vivax* infections. Radical cure to treat the dormant liver stages may enable larger reductions in *P. vivax* transmission. The results support the great potential of molecular surveillance in complementing traditional malariometric approaches.

## Introduction

Despite significant progress in reducing the burden of malaria in the Asia-Pacific over the last decade, recent World Malaria Reports have shown that these gains are not universal. And, where they have occurred, they are associated with an increase in the proportion of malaria due to *P. vivax* [1]. The differential impact of enhanced malaria control activities can be explained, in part, by fundamental biological and epidemiological differences between *P. vivax* and *P. falciparum*, including the ability of *P. vivax* to form dormant liver stages (hypnozoites) and a greater prevalence of low-density infections [2].

Assessment of malaria control interventions through conventional malariometric surveillance focuses on quantifying case numbers and parasite prevalence from which transmission intensity can be inferred. Case surveillance and cross-sectional surveys require comprehensive collection of data on parasitised individuals that is restricted by logistical constraints and the inability to detect very low-density infections. Furthermore, case surveillance has limited ability to identify subtle changes in parasite populations associated with changing epidemiology and selective pressures on the parasites [3].

Genotyping has been proposed as a complementary tool to identify early changes in parasite population structure, gene flow and parasite diversity [2, 4–8]. However, few molecular studies have assessed longitudinal molecular changes in the parasite, and none have compared co-endemic *P. falciparum* and *P. vivax* populations over a period longer than four years [9–13]. The lack of a comprehensive longitudinal evaluation of co-endemic parasite populations

is a major gap in our understanding of how public health interventions, such as the implementation of new antimalarial drug policies, differentially affect *P. vivax* and *P. falciparum* [4].

In 2004, clinical trials undertaken in Papua Indonesia demonstrated that both *P. falciparum* and *P. vivax* were highly resistant to sulphadoxine pyrimethamine and chloroquine, which were the first-line treatments against uncomplicated malaria at the time [14]. In March 2006, antimalarial guidelines were changed, and a universal Artemisinin-based Combination Therapy (ACT) policy was implemented. DHA-piperaquine (DP) was advised for uncomplicated malaria and intravenous artesunate for severe malaria. The change in policy to highly efficacious antimalarial treatment was associated with a 51% fall in the incidence of *P. falciparum* and a 28% fall in the incidence of *P. vivax* [1]. The aim of the current study was to characterize the temporal changes in the genetic diversity and structure of co-endemic *P. vivax* and *P. falciparum* populations over the course of more than a decade (2004–2017) of concerted interventions, including the introduction of a new treatment regimen into an area with multidrug resistant malaria.

## Results

Between 2004 and 2017, a total of 1,197 *P. vivax* and 1,566 *P. falciparum* clinical isolates were collected from patients with uncomplicated malaria, of which 628 (52%) *P. vivax* isolates and 671 (43%) *P. falciparum* isolates were available for molecular analysis. A total of 615 (97.9%) *P. vivax* and 666 (99.3%) *P. falciparum* isolates could be genotyped successfully (S1 Fig). The sample size ranged from 93 to 176 in each of the five predefined time intervals: pre-ACT-Policy change (2004–2006), early transition to ACT-Policy (2006–2009), late transition to ACT-Policy implementation (2009–2012), early Post-ACT implementation (2012–2015), and late Post-ACT implementation (2015–2017) (S1 Table). All markers exhibited a minimum 5% minor allele frequency, and a genotyping success rate exceeding >80% (S2 Table).

The parasitaemia, age and gender composition of the successfully genotyped samples were comparable to all samples available during each period (S1 Table). Complete demographic data were available for 96% (588/615) individuals infected with *P. vivax* and 96% (638/666) with *P. falciparum* (S3 Table). Age composition was comparable among the five periods for both species (S3 Table). The isolates were collected predominantly from adult patients (aged ≥15 years), contributing 76% (450) of *P. vivax* isolates and 86% (548) of *P. falciparum* isolates. Likewise, gender distribution was similar across the five periods for both species (S3 Table), with males comprising approximately half of all patients with *P. vivax* (45%, 278/588) and *P. falciparum* infections (49%, 328/640) (S3 Table).

The geometric mean parasitaemia differed significantly over the time intervals for both *P. vivax* ($p<0.001$) and *P. falciparum* infections ($p<0.001$; S3 Table). There was a trend of increasing *P. falciparum* parasitaemia over the study period, rising from 10,423 parasites/μL [95%CI 8,750–124,164] in 2004–2006 to 18,981 parasites/μL [95%CI 15,856–22,722] in 2015–2017 ($rho = 0.196$, $p<0.001$). However, there was no correlation between parasitaemia and multiplicity of infection (MOI) for either species (*P. vivax*: rho = -0.077, $p = 0.877$; *P. falciparum*: rho = -0.058, $p = 0.148$).

The proportion of polyclonal infections decreased significantly over time for both species. Polyclonal *P. vivax* infections fell from 67% (80/119) in 2004–2006 to 35% (33/93) in 2015–2017; ($p < 0.001$), while polyclonal *P. falciparum* infections fell from 28% (38/135) in 2004–2006 to 18% (22/125) in 2015–2017; ($p = 0.009$); Table 1). There was also a decrease in the complexity of infection over time in both the *P. vivax* and *P. falciparum* populations (S2 Fig). In the *P. vivax* population, the mean (SD) multiplicity of infection (MOI) decreased from 1.9 (0.7) in 2004–2006 to 1.4 (0.7) in 2015–2017 ($p<0.001$). In the *P. falciparum* population, the

**Table 1. Within-host and population diversity.**

| Period | N | Polyclonal N [%; CI95%] | MOI Mean (SD) | MOI Median (Max) | MLOCI Median (Max) | $H_E$ Mean (SD) | Rs Mean (SD) |
|---|---|---|---|---|---|---|---|
| **P. vivax** | | | | | | | |
| 2004–2006 | 119 | 80 [67; 59–76] | 1.9 (0.7) | 2 (4) | 3 (7) | 0.864 (0.06) | 14.6 (6.2) |
| 2006–2009 | 143 | 81 [57; 49–65] | 1.8 (0.8) | 2 (4) | 3 (7) | 0.858 (0.06) | 15.6 (6.5) |
| 2009–2012 | 114 | 44 [38; 29–47] | 1.4 (0.6) | 1 (4) | 2 (7) | 0.852 (0.07) | 16.2 (7.2) |
| 2012–2015 | 146 | 58 [40; 32–48] | 1.4 (0.6) | 1 (4) | 2 (7) | 0.854 (0.06) | 15.8 (6.1) |
| 2015–2017 | 93 | 33 [35; 26–45] | 1.4 (0.7) | 1 (5) | 1 (5) | 0.860 (0.07) | 17.0 (8.3) |
| **P. falciparum** | | | | | | | |
| 2004–2006 | 135 | 38 [28; 21–36] | 1.3 (0.5) | 1 (3) | 1 (5) | 0.594 (0.2) | 7.3 (2.5) |
| 2006–2009 | 128 | 38 [29; 21–37] | 1.3 (0.5) | 1 (3) | 2 (5) | 0.628 (0.3) | 7.5 (2.5) |
| 2009–2012 | 102 | 28 [27; 19–36] | 1.3 (0.4) | 1 (2) | 1 (4) | 0.623 (0.3) | 6.4 (2.2) |
| 2012–2015 | 176 | 35 [20; 14–26] | 1.2 (0.4) | 1 (2) | 1 (5) | 0.545 (0.3) | 5.2 (2.1) |
| 2015–2017 | 125 | 22 [18; 11–24] | 1.2 (0.4) | 1 (3) | 1 (5) | 0.602 (0.2) | 7.0 (2.3) |

MOI: Multiplicity of Infection; MLOCI: Number of multiallelic loci; CI95%: 95% Confidence interval of the proportion of polyclonal infections Rs: Allelic richness; $H_E$: Expected heterozygosity

corresponding change was 1.3 (0.5) to 1.2 (0.4) (*p* = 0.046; Table 1). The median (range) number of multiallelic loci per infection (MLOCI) fell in the *P. vivax* population (from 3 (1–7) to 1 (1–5); *p* = 0.005) but remained low in the *P. falciparum* population throughout the study period (1 (1–5) in 2004–2006 and 1 (1–5) in 2015–2017 (*p* = 0.161; Fig 1).

There was no temporal trend in genetic diversity, as measured by allelic richness (*Rs*), with moderate fluctuations observed in both the *P. vivax* and *P. falciparum* populations (Table 1). Over the study period, there was a slight increase in *Rs* in the *P. vivax* population from a mean (SD) of 14.6 (6.2) to 17 (8.3), but this was not significant (*p* = 0.999). In the *P. falciparum* population, *Rs* was 7.3 (2.5) in 2004–2006 and 7.0 (2.3) in 2015–2017 (*p* = 0.518). Similar trends were observed for the expected heterozygosity ($H_E$) in both species (Table 1).

Multi-locus genotypes (MLGs) were assembled from 636 *P. falciparum* isolates and 461 *P. vivax* isolates with complete genotyping data. In the *P. falciparum* population, 69 MLGs were multiply observed (repeated MLGs) among 206 individuals, and the proportion of these individuals increased from 32% (40/126) in 2004–2006 to 45% (54/119) in 2015–2017; (*p*<0.001; Table 2). In the *P. vivax* population, only 4 repeated MLGs were observed among 8 individuals; however, the proportion of these individuals also increased over time, from 0% during 2004–2006 to 7.7% (6/78) in 2015–2017 (*p* = 0.004; Table 2). Two of the 69 *P. falciparum* repeated MLGs persisted for up to 9 years (n = 15; Fig 2), while none of the four *P. vivax* repeated MLGs persisted for more than 3 months (n = 8; Fig 2, Table 2).

The multi-locus linkage disequilibrium (LD) in the *P. vivax* population was low throughout the study period, although the index of association ($I_A{}^S$) increased 2.2-fold from 0.0046 in 2004–2006 to 0.0102 in 2015–2017 (*p*<0.01) (Table 3). The LD in the *P. falciparum* population was consistently higher than the LD in *P. vivax*, and the index of association increased 5.6-fold between 2004–2006 and 2015–2017, from 0.0415 to 0.2340 (*p*<0.01). The trends of increasing LD over time remained after restricting the analysis to low complexity infections, confirming that the results were not affected by potential MLG reconstruction errors (Table 3). There was no evidence of a clonal outbreak in the *P. falciparum* population (S3 Fig).

The Bayesian clustering algorithm implemented in STRUCTURE software was unable to detect population substructure among the *P. vivax* isolates analysed (S4 Fig). In contrast, *delta K* analysis predicted between 2 and 4 *P. falciparum* subpopulations (S4 Fig). When assuming 2

## a. *Plasmodium vivax*

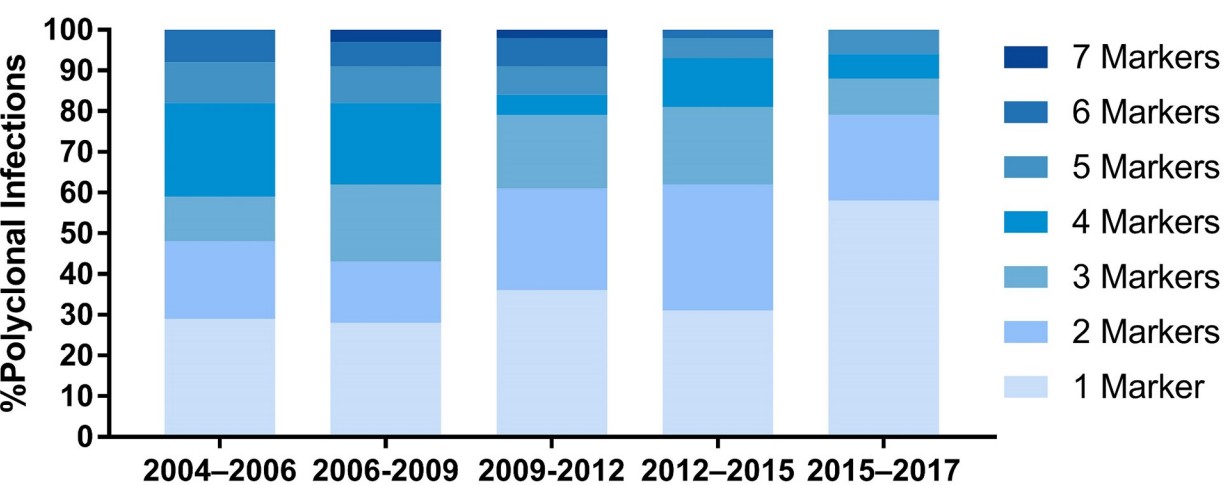

## b. *Plasmodium falciparum*

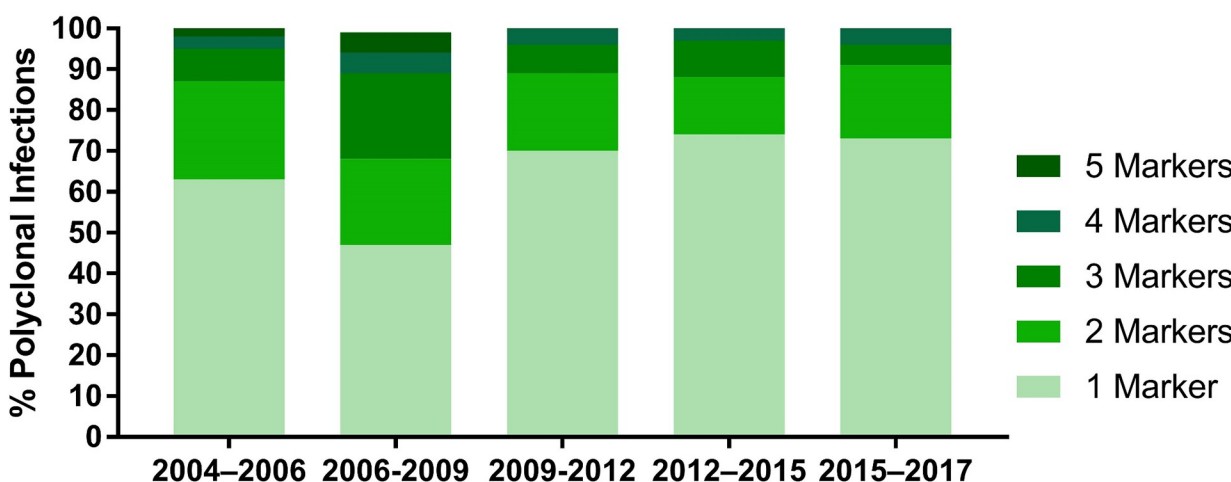

**Fig 1. Proportion of multiallelic loci per infection (MLOCI) by temporal period.** Bar charts illustrating the percentage of polyclonal infections with the given number of multiallelic loci for each of the 5 temporal periods in a) *P. vivax* and b) *P. falciparum*. Both species exhibit an overall decline over time in the percentage of infections with 2 or more multiallelic loci.

subpopulations, 70% (n = 466) of the isolates showed predominant (i.e., not mixed) ancestry to one of the two subpopulations (Fig 3A). Most of the temporal periods had a 3:2 ratio composition of isolates belonging to K1 or K2, respectively (S4 Table). When assuming four subpopulations, 50% (n = 337) of the isolates showed predominant ancestry to one of the four subpopulations. Amongst these non-mixed isolates (n = 337), in the first two temporal periods (2004–2006 and 2006–2009), 93% (57/61) and 76% (53/70) of the isolates had ancestry to either the K1 or K3 subpopulations (S4 Table). In contrast, in the late post-ACT transition period (2015–2017), 78% (n = 67/86) of the isolates had ancestry to either the K2 or K4 subpopulations (S4 Table). Notably, all isolates collected at the end of 2016 and throughout 2017 had ancestry to the minor K2 subpopulation (Fig 3B).

The pattern of sub-structure in the *P. falciparum* clinical isolates mirrored patterns observed in a previous study conducted in Papua between 2011 and 2014 [15]. Using

**Table 2. Frequency of infections with repeated multi-locus genotypes (MLGs).**

| Periods | *P. vivax* | | *P. falciparum* | |
|---|---|---|---|---|
| | Total infections with MLGs[a]; n | Proportion of infections with repeated MLGs; %, [CI95][b] | Total infections with MLGs[a]; n | Proportion of infections with repeated MLGs; %, [CI95][b] |
| 2004–2006 | 96 | 0 [0–0] | 126 | 32 [24–40] |
| 2006–2009 | 125 | 2 [1–4] | 124 | 18 [11–24] |
| 2009–2012 | 69 | 0 [0–0] | 98 | 28 [19–36] |
| 2012–2015 | 93 | 0 [0–0] | 169 | 37 [30–45] |
| 2015–2017 | 78 | 8 [2–14] | 119 | 45 [36–54] |
| Total | 461 | 1.7 [0.5–2.9] | 636 | 32 [29–36] |

a: Total number of infections with complete multi-locus genotypes (MLGs)

b. Proportion of individuals infected with repeated MLGs and corresponding 95% Confidence interval (CI95).

genotyping data generated on symptomatic and asymptomatic *P. falciparum* cases from Papua and three other regions of Indonesia (Bangka Belitung, West Kalimantan and Nusa Tenggara), the 2011–14 study found a notable sub-population of asymptomatic *P. falciparum* cases presenting in 2013 (defined as Papua asymptomatic K1) that appeared to have been imported from a region close to Nusa Tenggara [15]. We sought to determine whether the Papuan symptomatic K2 subpopulation observed here and the previously described asymptomatic K1 subpopulation reflected the same reservoir. Multiple correspondence analysis (MCA) on the current and previously described *P. falciparum* datasets [15, 16] revealed higher genetic relatedness between the Papuan symptomatic K2 subpopulation, the putatively imported Papuan asymptomatic K1 subpopulation and the infections from Nusa Tenggara than the other Papuan infections (S5 Fig).

## Discussion

This study presents a comprehensive longitudinal genetic investigation of *P. falciparum* and *P. vivax*, comprising data from over 1,200 parasite isolates collected over 14 years. It is the first longitudinal genetic analysis documenting the diversity and structure of co-endemic *P. falciparum* and *P. vivax* populations before, during, and after the implementation of a universal ACT policy. The results reveal important molecular cues consistent with differential patterns in the decline in transmission of *P. vivax* and *P. falciparum* following policy change in a region with multidrug resistant malaria; these findings have been confirmed with complementary epidemiological data from a large-scale case surveillance study in the same area [1, 15]. The genetic results also highlight the emergence of a subpopulation of potentially adaptive clinical *P. falciparum* infections in the late post-ACT transition period.

Multiple clone infections can arise by superinfection in the mosquito (e.g. due to interrupted feeding) as well as superinfection in the patient following serial infected mosquito-bites. The risk of superinfection is likely to be greater in high transmission settings. Previous studies have demonstrated a positive correlation between the complexity and/or proportion of polyclonal malaria infections and transmission intensity [17–19]. Consequently, reduction in the complexity or prevalence of polyclonal malaria infections has been proposed as an early marker of decreasing transmission in the given population [18]. Our study revealed significant

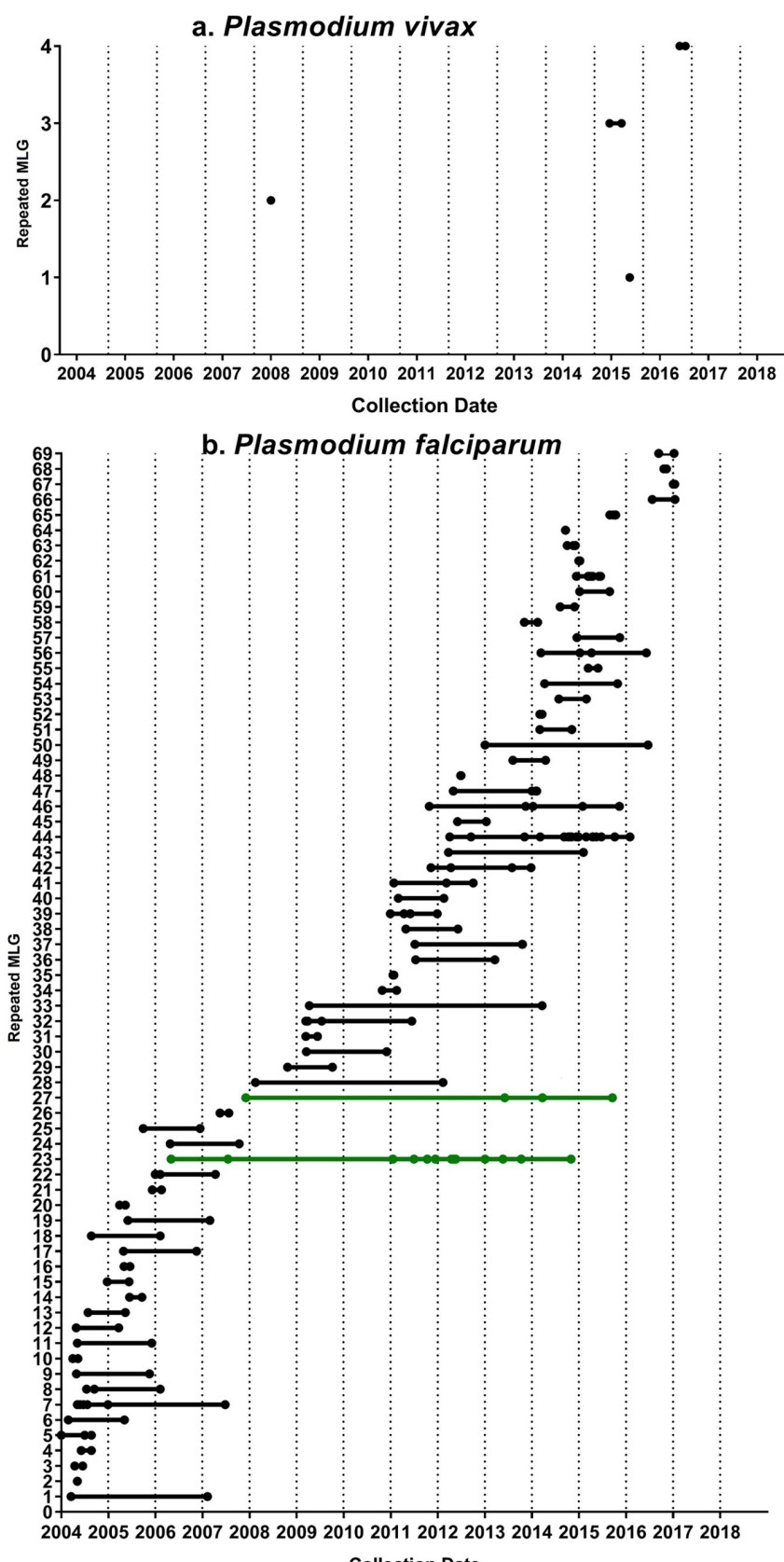

**Fig 2. Persistence of repeated MLGs over time.** Dot points illustrating the year when repeated MLGs were detected in each of a) *P. vivax* and b) *P. falciparum*. The *P. vivax* repeated MLGs were constructed across 8 loci, and the *P. falciparum* infections were constructed across 9 loci. The persistence of *P. falciparum* strains (repeated MLGs) reached up to 9 years (green) and was markedly greater than for *P. vivax*, which did not persist for over a year. However, most *P. falciparum* strains (repeated MLGs) had shorter duration (less than a year).

reductions in the proportion of polyclonal infections in *P. vivax* and *P. falciparum* between the earliest and latest ACT transition periods (1.8- and 1.6-fold, respectively). The difference in the magnitude of the reduction in polyclonal infections between the species likely represents the higher pre-ACT baseline prevalence of polyclonal *P. vivax* infections compared to *P. falciparum*. These findings highlight constraints in the utility of the complexity or prevalence of polyclonal infection to monitor reductions in malaria transmission in low endemic settings where the complexity is low at the start of monitoring. Although not observed here, it should also be noted that some studies have observed high complexity of infection in low endemic settings, potentially reflecting factors such as polyclonal imported infections [20].

In *P. falciparum*, population genetic diversity has been proposed to correlate positively with endemicity [17]. Although our study found a trend of declining allelic richness in the *P. falciparum* population from 2004 until 2015, this did not reach statistical significance. Indeed, Nkhoma and colleagues also found population diversity to be limited as a measure of endemicity in their longitudinal survey of *P. falciparum* on the Thai-Myanmar border [18]. This finding may reflect the high human movement between Papua and other Indonesian islands, which could provide a diverse reservoir of new alleles being introduced into Papua [14, 15]. A study of *P. vivax* population diversity in Sri Lanka reported increasing diversity despite declining transmission, potentially reflecting imported cases amongst other factors. [20]. The available evidence suggests that low *P. vivax* diversity may not be a prerequisite for elimination of this species. Indeed, in contrast to *P. falciparum*, there was no change in genetic diversity in the Papuan *P. vivax* population over time, likely reflecting a combination of importation and enhanced transmission opportunities afforded by the dormant liver stage.

**Table 3. Multi-locus Linkage Disequilibrium.**

| Subgroups | All infections, N | All infections, $I_A^S$ | Low complexity, N | Low complexity, $I_A^S$ |
|---|---|---|---|---|
| **P. vivax** | | | | |
| 2004–2006 | 96 | 0.0046* | 51 | 0.0064[NS] |
| 2006–2009 | 125 | 0.0038[NS] | 74 | 0.0036[NS] |
| 2009–2012 | 70 | 0.0116** | 52 | 0.0113* |
| 2012–2015 | 92 | -0.0043[NS] | 65 | -0.0092[NS] |
| 2015–2017 | 78 | 0.0102** | 66 | 0.0157* |
| **P. falciparum** | | | | |
| 2004–2006 | 126 | 0.0415** | 116 | 0.0452** |
| 2006–2009 | 124 | 0.0433** | 106 | 0.0461** |
| 2009–2012 | 98 | 0.0504** | 91 | 0.0527** |
| 2012–2015 | 169 | 0.0375** | 161 | 0.0381** |
| 2015–2017 | 119 | 0.234** | 113 | 0.2257** |

Only samples with no missing data were included in the analyses.

* $p < 0.05$

** $p < 0.01$

NS: not significant.

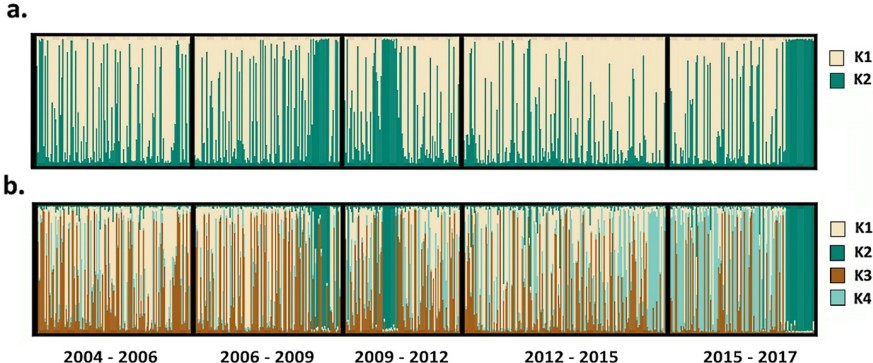

**Fig 3. Temporal trends in the prevalence of *P. falciparum* sub-populations.** STRUCTURE bar plots illustrating the distribution of *P. falciparum* isolates with ancestry to the given K sub-populations over time. Panel a) presents the data assuming K = 2, and panel b) presents the data assuming K = 4. Each vertical bar presents a single isolate, whose relative ancestry to each of the given K sub-populations is illustrated by the proportionate colour-coded segments. Isolates are ordered from left to right on the x-axis by date of collection (oldest to most recent). At K = 2, each temporal period exhibits an approximate 3:2 ratio composition of isolates with predominant ancestry (>85%) to K1 and K2 respectively. At K = 4, majority of isolates in the first two temporal periods have predominant ancestry to K1 or K3, whilst the majority in the later periods have predominant ancestry to K2 or K4. Isolates with predominant ancestry to K2 prevail in late 2016 and throughout 2017.

Increasing proportions of individuals harbouring repeated multi-locus genotypes (MLGs) and long persistence of repeated MLGs are strong predictors of declining malaria transmission [18, 21, 22]. Although only 4 *P. vivax* repeated MLGs were detected in the study, there was a modest increase in their prevalence in the late temporal periods, suggesting a discrete increase in inbreeding events in this species. A total of 69 repeated MLGs were detected in the *P. falciparum* population, and their proportions demonstrated a significant increase (from 32 to 45%) between the pre-ACT and late post-ACT transition periods, providing strong evidence of increasing self-fertilization events over time in this species. Strikingly, two of the *P. falciparum* repeated MLGs (3%) persisted for 8 and 9 years. A previous longitudinal study of *P. falciparum* conducted in a low-endemic area demonstrated a similar distribution in repeated MLGs prevalence over time, and also observed persistence of strains for up to 8 years [22].

Since mixed-clone infections are a main contributor of cross-fertilisation events, LD is expected to correlate negatively with the proportion of polyclonal infections and thus, theoretically, should increase as the transmission intensity decreases in the absence of outbreaks. There was no evidence of large outbreaks of one or a few genetic strains in the *P. vivax* or *P. falciparum* populations in our study. Although overall LD levels remained low, there was a modest (2.2-fold) increment in $I_A^S$ estimates in the *P. vivax* population between the pre-ACT and late post-ACT transition periods, likely reflecting the decline in polyclonal infections and subtle increase in frequency of repeated MLGs. In *P. falciparum*, a larger (5.6-fold) increase in $I_A^S$ estimates was observed between the pre-ACT and late post-ACT period. In conjunction with the increasing prevalence of repeated MLGs and long persistence of strains, the LD results suggest a substantial decline in cross-fertilisation over time in the *P. falciparum* population.

Together, the molecular data on complexity and prevalence of polyclonal infections, repeated MLGs prevalence and LD, suggest declining transmission and increasing inbreeding over time in *P. falciparum* and, to a lesser extent, in *P. vivax*. These results are in line with epidemiological surveillance data collected from 2004 to 2009 [1], which also found larger reductions in the incidence of *P. falciparum* cases (51%), than in *P. vivax* cases (28%) [1]. These findings highlight the utility of molecular data on complexity or prevalence of polyclonal

infections, repeated MLGs prevalence or LD to characterise long-term changes in parasite transmission intensity in regions with comparable endemicity to Mimika [8].

The change in treatment policy to ACT as first-line treatment for all species of malaria, intense vector control, and annual bed net distribution (implemented since 2004), may have all contributed to the epidemiological and genetic changes observed in the *P. vivax* and *P. falciparum* populations in Mimika. However, we hypothesise that the implementation of ACT had the greatest impact [23]. Artemisinins clear parasitaemia faster than less potent drugs such as chloroquine and have demonstrated potency against the transmissible gametocyte stages. Therefore, in regions where they remain effective, ACTs have a greater impact in reducing parasite biomass and overall transmission than other conventional drugs [24]. Vector control has been shown to be effective in reducing transmission in Papua New Guinea [25]. However, given the bionomics of the local *Anopheles* species in Mimika (mostly exophilic behaviour), it is likely that bed-nets may have had little impact on malaria in this population [1, 14].

In addition to the molecular cues of declining transmission intensity, parasite genotyping revealed temporal changes in the frequency of four *P. falciparum* sub-populations detected in the study. The most notable change was the fluctuation in prevalence of a divergent subpopulation, defined as K2, which was observed in small clusters in 2009 and 2011, before re-emerging and predominating towards the end of the study period (late 2016 and throughout 2017). It remains unclear whether the K2 subpopulation was introduced from elsewhere in Indonesia or overseas, or emerged locally, or whether its recent expansion was neutral or driven by favourable selective pressures from antimalarial drugs or other forces. Cluster-based analyses demonstrated that the K2 sub-population was genetically closely related to isolates from Nusa Tengarra, suggestive of importation from this province or a nearby region. However, as the microsatellite markers may be limited in their ability to determine geographic origin, we cannot exclude the possibility that the divergent infections reflect local adaptations in response to the changing epidemiology. Indeed, the recent expansion of the K2 subpopulation is particularly interesting in the context of a recent genomic study, which revealed closer genetic relatedness between three artemisinin-resistant *P. falciparum* infections detected in Papua New Guinea with infections derived from Mimika than with other Papua New Guinean isolates [26]. Further temporal investigation of the *P. falciparum* infections in Mimika are needed with appropriate phenotypic data.

In summary, our study demonstrates that enhanced malaria control activities in a co-endemic setting had a significant impact on the local transmission dynamics of both *P. falciparum* and *P. vivax*. The greater genetic changes observed in the *P. falciparum* population likely reflect a parasite population more susceptible to schizontocidal antimalarial drugs whereas the lesser impact on the *P. vivax* population emphasises the need for the radical cure of this species. The recent emergence and predominance of a divergent *P. falciparum* subpopulation highlights the importance of surveillance to inform control programs of potential new threats.

## Materials and methods

### Ethics statement

Ethical approval for the study was obtained from the Eijkman Institute Research Ethics Commission, Eijkman Institute for Molecular Biology, Jakarta, Indonesia (EIREC-47, EIREC-67, and EIREC-75), the Ethics committee of the National Institute of Health Research and Development, Indonesian Ministry of Health, Jakarta, Indonesia (NIHRD: KS.01.01.6.591, NIHRD: KS.02.01.2.3.4579, NIHRD: KS.02.01.2.1.4042, NIHRD: KS.02.01.2.1.1615 and NIHRD: LB.03.02/KE/4099/2007), and the Human Research Ethics Committee of the Northern Territory (NT) Department of Health & Families and Menzies School of Health Research, Darwin,

Australia (MSHR: 02/55, MSHR: 07/06, MSHR: 03/64, MSHR: 05/16, MSHR: 07/14 and HREC 2010–1396).

## Study site

The study was conducted in Mimika District, located in the south of Papua Province, Indonesia (S6 Fig). Details on the epidemiology of the study site have been reported previously [1, 15]. Briefly, malaria transmission is perennial in Mimika, but almost exclusive to the lowlands. Most malaria infections are caused by *P. falciparum* and *P. vivax*, but *P. malariae* and *P. ovale* are also endemic. Papua Province has historically harboured high levels of antimalarial drug resistance in both *P. vivax* and *P. falciparum* [27]. Surveys conducted between 2004 and 2006 demonstrated 65% treatment failure against CQ monotherapy at day-28 for vivax malaria and 48% failure against CQ plus sulphadoxine-pyrimethamine (SP) for *P. falciparum* [14]. Epidemiological surveillance data collected from local health facilities and cross-sectional studies in Mimika District showed an overall decrease of malaria incidence assuming shifts in treatment-seeking behaviour, from 889 infections per 1,000 person-years in 2004–2006 to 522 in 2010–2013 [1]. The incidence of *P. falciparum* cases fell from 511 per 1,000 person-years in 2004–2006, to 249 in 2010–2013 and, the incidence of *P. vivax* cases fell from 331 to 239 per 1000 person-years over the same periods [1].

## Patient sampling framework

Samples were sourced from patients recruited to clinical and *ex vivo* surveillance studies carried out in Mimika between 2004 and 2017 [14, 28–32]. The same sampling strategy was applied throughout the study period and ensures a homogenous patient catchment areas and demographics. Briefly, blood samples were collected from consenting, symptomatic patients with uncomplicated malaria attending the Rumah Sakit Mitra Masyarakat (RSMM) hospital. Peripheral parasitaemia and species identity were determined by light microscopy examination of Giemsa-stained blood smears. Available samples were selected for parasite genotyping. Genomic DNA (gDNA) was extracted from either 2 mL of venous blood using the QIAamp DNA Midi Kit (Qiagen), or 100 μL of red blood cell pellet using the QIAamp DNA Mini Kit. Species confirmation was performed using a nested PCR protocol [33].

## Microsatellite typing

Nine short tandem repeat (STR) markers (*ARAII*, *PfPK2*, *poly-alpha*, *TA1*, *TA42*, *TA60*, *TA81*, *TA87* and *TA109*) described by Anderson *et al* were used to genotype *P. falciparum* isolates [34]. For *P. vivax*, a panel comprising eight STR markers (*MS1*, *MS5*, *MS10*, *MS12*, *MS20*, *MS16*, *msp1F3*, and *PV3.27*) described by Koepfli *et al* and Karunaweera *et al*. were used [35, 36]. The primers and PCR conditions for the assays are described elsewhere [37]. The labelled PCR products were sized on an ABI 3100 Genetic Analyser with GeneScan LIZ-600 size standard (Applied Biosystems). The resulting electrophoretograms were analysed using the online, open-access vivaxGEN platform [38]. All genotypes can be accessed in vivaxGEN. The *P. vivax* genotypes are available under the batch codes IDPV-XXV, IDPV-TES, IDPV-ACT and IDPV-ACT2, and the *P. falciparum* genotypes are available under IDPF-XXV, IDPF-TES, IDPF-ACT and IDPF- ACT2. An arbitrary intensity threshold of 100 relative fluorescence units (RFU) and minimal 33% peak intensity of minor relative to predominant peaks was used to reduce background noise/artefacts. Only samples with information in at least 50% of the loci were considered successfully genotyped.

## Data analysis

**Sample grouping.** Samples were grouped according to their collection date into 5 predefined periods: pre-ACT-Policy change (April 2004 to March 2006–24 months); early transition to ACT-Policy (April 2006 to March 2009–36 months); late transition to ACT-Policy implementation (April 2009 to March 2012); early Post-ACT implementation (April 2012 to March 2015–36 months); and late Post-ACT implementation (April 2015 to May 2017–24 months).

**Population genetic analysis.** Multiple parasite clones were defined if more than one allele at one or more loci was present in an individual sample. The MOI was defined as the maximum number of alleles at any locus for a given sample. The number of MLOCI was also estimated for closer inspection of the complexity of individual infections in the effort to identify any subtle changes in transmission patterns over time.

Temporal analysis of the genetic relatedness between infections was conducted by assessment of the proportion of shared alleles, frequency and duration of repeated MLGs, and the proportion of repeated MLGs per period and illustrated with neighbour-joining trees generated using the ape package in R [39]. Isolates without missing data were used to build MLGs from the predominant allele at each locus. The frequency and temporal duration of repeated MLGs was estimated using the R packages adegenet and RClone [40].

LD, which is the non-random association of alleles at different loci, was measured using the standardised index of association ($I_A^S$). $I_A^S$ compares the observed variance of the number of mismatched loci between haplotypes to the expected variance if the loci were randomly associated.[21]. The web-based LIAN 3.5 software was used to calculate the estimates [41]. Briefly, multi-locus LD was compared between groups in search of evidence of increasing LD. Ten thousand permutations of the data was used to assess the significance of the estimates. LD analysis was performed on all MLGs and using low complexity infections (maximum of 1 multi-allelic locus) only.

Population genetic diversity was estimated using the allelic richness (*Rs*), a measure of the number of alleles at a given locus with normalisation (rarefaction) for sample size. *Rs* was calculated using the hierfstat package in R [42]. Measures of the expected heterozygosity ($H_E$) were also provided for comparison with previous studies.

Population structure was assessed using STRUCTURE software version 2.3.3 [43]. The simulation was run using 20 replicates, with 100,000 burn-in and 100,000 post burn-in iterations for each estimate of K (number of sub-populations), ranging from 1–10. The model parameters included admixture with correlated allele frequencies. The *delta K* method was used to derive the most probable K, implemented with STRUCTURE HARVESTER [44, 45]. An arbitrary threshold of 85%> was used to define ancestry to the different K subgroups. Distruct software version 1.1 was used to display the results from STRUCTURE as bar plots [46].

**Statistical tests.** SPSS software (version 24) was used for statistical analysis. Differences in MOI, percentage of polyclonal infections, proportion of infections with multiply observed MLGs, allelic richness (*Rs*), and expected heterozygosity ($H_E$) between subgroups and species were assessed using the Mann-Whitney U or Kruskal-Wallis test, spearman correlation for continuous trends and chi-square test for trends and differences in proportion.

## Supporting information

**S1 Table. Demographic data by temporal period for *P. falciparum* and *P. vivax* isolates included in the study versus all cases screened. GM: Geometric mean; 95%CI: 95% Confidence interval; # Superscript indicate number of missing data for Age, Sex and Parasitaemia.**
(DOC)

**S2 Table. Marker diversity and genotyping success rate in *P. vivax* and *P. falciparum*.**
(DOC)

**S3 Table. Demographic data by temporal period for *P. vivax* and *P. falciparum* isolates included in the study.** GM: Geometric mean; 95%CI: 95% Confidence interval.
(DOC)

**S4 Table. Ancestry of *P. falciparum* isolates assuming 2 and 4 sub-populations.**
(DOC)

**S1 Fig. Flowchart illustrating the sample selection process.**
(TIFF)

**S2 Fig. Mean multiplicity of Infection (MOI) over time in *P. vivax* and *P. falciparum*.**
(TIFF)

**S3 Fig. Neighbour-joining plots illustrating the genetic diversity in the *P. falciparum* populations across the five periods.**
(TIFF)

**S4 Fig. STRUCTURE results in *P. vivax* and *P. falciparum*.** Panel a) provides a STRUCTURE bar plot constructed from the *P. vivax* data at K = 2, illustrating a lack of notable substructure. Panel b) provides a scatter plot of K against Delta K for the *P. falciparum* data, illustrating peaks at K = 2 and K = 4.
(TIFF)

**S5 Fig. Multiple Correspondence Analysis plot illustrating the relatedness between the Papuan *P. falciparum* subpopulations (K1 and K2) and other Indonesian Islands.** Higher genetic relatedness was observed between the putatively imported Papuan asymptomatic K1 subpopulation (Papua Asymp K1, green circles) [15], the Papuan symptomatic K2 subpopulation (Papua Symp K2, red circles), and the infections from Nusa Tenggara Timur (aquamarine circles)[16] than the other Papuan symptomatic infections from the current study (Papua Symp Other, pink circles).
(TIFF)

**S6 Fig. Map illustrating the location of the study site.** Map adapted from Kenangalem et al. 2019 illustrating the location of Mimika District within Papua Province, Indonesia.
(TIFF)

## Acknowledgments

We would like to thank the patients who contributed their samples to the study and the health workers and field teams who assisted with the sample collections.

## Author Contributions

**Conceptualization:** Sarah Auburn.

**Data curation:** Hidayat Trimarsanto, Ric N. Price, Jutta Marfurt.

**Formal analysis:** Zuleima Pava, Sarah Auburn.

**Funding acquisition:** Nicholas M. Anstey, Ric N. Price.

**Investigation:** Zuleima Pava, Agatha M. Puspitasari, Angela Rumaseb, Irene Handayuni, Retno A. S. Utami, Yusrifar K. Tirta, Grennady Wirjanata, Steven Kho.

**Project administration:** Jutta Marfurt, Sarah Auburn.

**Resources:** Hidayat Trimarsanto, Nicholas M. Anstey, Jutta Marfurt, Sarah Auburn.

**Supervision:** Leily Trianty, Faustina Burdam, Enny Kenangalem, Jeanne Rini Poespoprodjo, Rintis Noviyanti, Ric N. Price, Jutta Marfurt, Sarah Auburn.

**Validation:** Zuleima Pava, Agatha M. Puspitasari, Angela Rumaseb.

**Visualization:** Zuleima Pava, Sarah Auburn.

**Writing – original draft:** Zuleima Pava, Sarah Auburn.

**Writing – review & editing:** Zuleima Pava, Faustina Burdam, Enny Kenangalem, Nicholas M. Anstey, Jeanne Rini Poespoprodjo, Rintis Noviyanti, Ric N. Price, Jutta Marfurt, Sarah Auburn.

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
