## [Decision Letter · Decision Letter 0]

5 Feb 2020

Dear Dr Auburn,

Thank you very much for submitting your manuscript "Longitudinal molecular surveillance confirms interruption of P. vivax and P. falciparum transmission following implementation of a universal policy of Artemisinin-based Combination Therapy in Papua, Indonesia" for consideration at PLOS Neglected Tropical Diseases. As with all papers reviewed by the journal, your manuscript was reviewed by members of the editorial board and by several independent reviewers. In light of the reviews (below this email), we would like to invite the resubmission of a significantly-revised version that takes into account the reviewers' comments. 

We cannot make any decision about publication until we have seen the revised manuscript and your response to the reviewers' comments. Your revised manuscript is also likely to be sent to reviewers for further evaluation.

Sincerely,

Gregory Deye

Guest Editor

Walderez Dutra

Deputy Editor

Reviewer's Responses to Questions

**Key Review Criteria Required for Acceptance?**

**Methods**

-Are the objectives of the study clearly articulated with a clear testable hypothesis stated?

-Is the study design appropriate to address the stated objectives?

-Is the population clearly described and appropriate for the hypothesis being tested?

-Is the sample size sufficient to ensure adequate power to address the hypothesis being tested?

-Were correct statistical analysis used to support conclusions?

-Are there concerns about ethical or regulatory requirements being met?

Reviewer #1: Methods are mostly adequate for the aims of the study. I note, however, that a population bottleneck is mentioned (e.g., Author Summary and Discussion, line 292) but no formal test for bottleneck was applied to the samples. A LD test was correctly used, but described in a rather weird way. In fact, the standardised index of association implemented in LIAN software does not "compare the observed variance in the numbers of alleles shared between parasites with that expected when parasites share no alleles at different loci", as stated in lines 393-395. In fact, the index compares the observed variance of the number of alleles at which each pair of haplotypes differ in the population (i.e., the allele mismatch distribution) with the variance expected under random association of alleles. This is very important, because parasites may share identical alleles in a panmictic population -- therefore, the null hypothesis does not imply "no alleles shared"!

**Results**

-Does the analysis presented match the analysis plan?

-Are the results clearly and completely presented?

-Are the figures (Tables, Images) of sufficient quality for clarity?

Reviewer #1: Overall, results are well presented in the main text. Tables are clear and serve the purpose of presenting the main findings. 

Figure 1 is ok, but may be misleading; a pie chart would possibly be more adequate for the purpose of showing the, among multiple-clone infections, the proportion of those with 1 or more alleles found at a single or a few loci tends to increase with time. The bar chart may give the impression of an increasing overall prevalence of multiple-clone infections, which is clearly not true. 

Figure 2 is not very effective in presenting the results. A new figure inspired in Figures 1 and 3 of reference 9, for example, can be a better option. The point here is to show that particular haplotypes shared by two or more isolates persist over time.

**Conclusions**

-Are the conclusions supported by the data presented?

-Are the limitations of analysis clearly described?

-Do the authors discuss how these data can be helpful to advance our understanding of the topic under study?

-Is public health relevance addressed?

Reviewer #1: Most conclusions are supported by the data, but a few important exceptions must be mentioned. First, the title: "Longitudinal molecular surveillance confirms interruption ...". Overall, the article is nicely written, the sample size is large enough for the proposed analysis, results are clearly presented... but the title is completely misleading. Data is this paper does not "confirm" that malaria transmission has been interrupted in Papua! Such a "confirmation" would require evidence that all remaining cases are imported, not locally transmitted, and no data support this! I would suggest that the data are consistent with a decrease in malaria transmission (mostly that of P. falciparum) following the implementation of the universal ACT policy. That is all.

What is consistent with transmission decline? Essentially, the decrease in the prevalence of multiple-clone infections and in average multiplicity of infection over time. No change in genetic diversity was observed during the study period. This is exactly what the study in reference 17 has shown along the Thai-Myanmar border. Interestingly, no change in genetic diversity of parasites (and no evidence of bottleneck) was found in another setting, Sri Lanka, with a much more drastic decrease in malaria transmission (doi: 10.1017/S0031182013002278, not cited in the text). 

Is increased LD consistent with decreased transmission. Not necessarily. LD is a consequence of reduced recombination and, of course, any factor reducing recombination will affect LD estimates. For example, malaria incidence may increase due to a clonal outbreak -- with increased transmission and increased LD. Therefore, it is important to document changes in LD over time, but they do not "confirm" that malaria transmission has declined. 

I have previously mentioned that the study does not provide formal evidence for a population bottleneck in parasite populations. (I would guess that, even if properly tested, the results would still be negative.) However, even if the authors found evidence for bottleneck, this does not necessarily "confirm" that transmission has been interrupted or even decreased. A selective sweep induced by the widespread of a new drug, for example, might lead to genetic changes at the population level that are consistent with a bottleneck, even if transmission levels have not been greatly affected.

Finally, there is throughout the paper a relatively loose use of the term "structure". If there is significant LD, of course malaria parasites are structured into lineages or subpopulations that do not recombine as much as expected for a randomly mating population. Therefore, stating that "there was no population structure among the P. vivax isolates (line 207) is simply incorrect. One can cautiously say that the Bayesian clustering algorithm implemented in STRUCTURE software was unable to detect population structure in the sample analysed. Whether STRUCTURE is the best strategy for detecting "structure" in populations at LD is debatable (see the STRUCTURE manual for a nice discussion on this topic).

The most likely cause of LD is the reduced proportion of multiple-clone infections documented during the study period, although it is biologically not true that "cross-fertilisation is only possible in mixed-clone infections" (as stated in line 280). Superinfection in the mosquito (due, e.g., to interrupted feeding) may also allow for cross-fertilisation. However, other factors may favour LD -- for example, if patient recruitment strategies have changed over time given the declining number of available patients, samples in more recent years may have been collected from a more (e.g., geographically) heterogeneous population where panmixia would be much less likely. These limitations must be recognised instead of overinterpreting LD results.

**Editorial and Data Presentation Modifications?**

Reviewer #1: (No Response)

**Summary and General Comments**

Reviewer #1: Overall, the paper is nicely written and describes important findings that are surely interesting to the broad audience of PLoS NTD. A better discussion of the data and, more specifically, their implications for malaria epidemiology, would render the paper even more interesting.

PLOS authors have the option to publish the peer review history of their article (what does this mean?). If published, this will include your full peer review and any attached files.

Reviewer #1: No
---

## [Decision Letter · Decision Letter 1]

3 Apr 2020

Dear Dr Auburn,

Thank you very much for submitting your manuscript "Longitudinal molecular surveillance confirms reduction of P. vivax and P. falciparum transmission following implementation of a universal policy of Artemisinin-based Combination Therapy in Papua, Indonesia" for consideration at PLOS Neglected Tropical Diseases. As with all papers reviewed by the journal, your manuscript was reviewed by members of the editorial board and by several independent reviewers. The reviewers appreciated the attention to an important topic. Based on the reviews, we are likely to accept this manuscript for publication, providing that you modify the manuscript according to the review recommendations. 

Sincerely,

Gregory Deye

Guest Editor

Walderez Dutra

Deputy Editor

Reviewer's Responses to Questions

**Key Review Criteria Required for Acceptance?**

**Methods**

-Are the objectives of the study clearly articulated with a clear testable hypothesis stated?

-Is the study design appropriate to address the stated objectives?

-Is the population clearly described and appropriate for the hypothesis being tested?

-Is the sample size sufficient to ensure adequate power to address the hypothesis being tested?

-Were correct statistical analysis used to support conclusions?

-Are there concerns about ethical or regulatory requirements being met?

Reviewer #2: (No Response)

Reviewer #3: The objectives of the study, to measure changes over time in population parameters in Plasmodium falciparum and Plasmodium vivax that may reflect changes in transmission, and to relate these to the introduction of artemisinin-based Combination Therapy (ACT) in Papua New Guinea, is testable and clearly stated. Statistical analyses used are appropriate. However, it’s not clear to me that the population bottleneck analyses add much. The comprehensive longitudinal sampling over 14 years of approximately 600 samples for each species is sufficient to address the hypothesis being tested.

**Results**

-Does the analysis presented match the analysis plan?

-Are the results clearly and completely presented?

-Are the figures (Tables, Images) of sufficient quality for clarity?

Reviewer #2: (No Response)

Reviewer #3: The results are clearly presented although Table 1 would be more appropriate as a Supplementary Table with a consolidated summary presented in the text. In Figure 3 the various colors are too similar to each other.

**Conclusions**

-Are the conclusions supported by the data presented?

-Are the limitations of analysis clearly described?

-Do the authors discuss how these data can be helpful to advance our understanding of the topic under study?

-Is public health relevance addressed?

Reviewer #2: (No Response)

Reviewer #3: The conclusions are in large part supported by the data. However, although the rationale for assuming a bottleneck may have occurred due to the introduction of a more effective drug therapy is reasonable, the analyses do not bear it out. Depending on the model used in the Bottleneck software either several bottlenecks are detected or none. By the authors own admission, the test of excess heterozygosity may not be sensitive enough to detect subtle changes. I would recommend removing bottleneck analyses altogether or using a more sensitive test if available.

**Editorial and Data Presentation Modifications?**

Reviewer #2: According to the journal policies, shouldn’t ‘Plasmodium’ be spelled out in the title?

The authors state that all data is available, and I assume it is saved in VivaxGEN. Please clarify.

Reviewer #3: (No Response)

**Summary and General Comments**

Reviewer #2: Main comment:

The authors argue that “Genetic epidemiology is gaining widespread interest as a tool that can enhance conventional malaria surveillance”, as obtaining classical metrics such as case numbers can be difficult. While I certainly agree, I am puzzled that the authors do make attempts to compare their genotyping data to incidence or prevalence data. They give some incidence data in the methods section (though these numbers appear not to sum up, see comment below). I imagine that for all periods of sampling test positivity rate or incidence for both species was collected. A figure comparing such number to genotyping indices would provide information on the additional benefits of genotyping.

Minor comments:

1) The title could be more informative, e.g. by changing to 

“Surveillance over 14 years confirms reduction of P. vivax and P. falciparum transmission following implementation of a universal policy of Artemisinin-based Combination Therapy in Papua, Indonesia”

“Longitudinal” is a broad term, and, for example, is often used for cohort studies of much shorter duration.

2) Line 49-50: The higher level of outbreeding has already been mentioned above in lines 42-43. No need to repeat it in the abstract.

3) Line 175: Does the word ‘bottlenecking’ really exist?

4) Instead of the term ‘Multi-locus genotype’, ‘haplotype’ is often used. Following that, I am not sure whether the term ‘moMLG’ should be introduced to the field of molecular epidemiology. ’Repeated haplotype’ would be an easier term.

5) Lines 263-265: It is correct that a reduction in complexity of infections has been proposed as a marker of declining transmission. Indeed, the study cited by Nkhoma et al found such a pattern, in a setting where the reduction in transmission was very pronounced (1-fold). However, other studies (from Africa, the South Pacific, and elsewhere) have found complexity to change slowly. Taken all available data into account, I would argue that high complexity even when transmission is reduced is a common pattern.

6) Lines 281-283: It could be mentioned that high Pv diversity was not only found during ‘declining transmission’, but essentially to the point of elimination. It appears low Pv diversity is not a prerequisite for elimination.

7) Lines 391-194: “an overall decrease of malaria incidence, from 406 infections per 1,000 person-years in 2004-2006 to 351 in 2010-2013 [1]. The incidence of P. falciparum cases fell from 511 per 1,000 person-years in 2004-2006, to 141 per 1,000 in 2010-2013 and, the incidence of P. vivax cases fell from 331 to 146 per 1000 person-years over the same periods [1].”

How can the overall decrease (which is the sum of Pf and Pv) be around 15%, when incidence of Pf fell 3-fold and incidence of Pv fell 2-fold? This seems impossible. The overall change should be some kind of average. Pf incidence alone is higher in 2004-2006 than overall incidence!

8) Discussion: In a previous study (reference 15) the authors have identified the K1 and K2 P. falciparum populations, and suggested importation along with other possibilities as explanation. I am surprised the current manuscript does not make any references to these hypotheses. Given their additional data, can the authors confirm or reject some the their previous hypotheses?

Reviewer #3: The authors are able to measure a number of alterations in population parameters for P. falciparum and P. vivax: a reduction in both the proportion and complexity of polyclonal infections, an increase in the proportion of moMLG in P. falciparum, sub-population structure in P. falciparum but not P. vivax, and no reduction in diversity for either. The results are of interest and have public health relevance as they highlight the differential reduction in transmission for the two species in a co-endemic setting.

PLOS authors have the option to publish the peer review history of their article (what does this mean?). If published, this will include your full peer review and any attached files.

Reviewer #2: No

Reviewer #3: No
---

## [Editor Report · Decision Letter 2]

15 Apr 2020

Dear Dr Auburn,

We are pleased to inform you that your manuscript 'Molecular surveillance over 14 years confirms reduction of Plasmodium vivax and falciparum transmission after implementation of Artemisinin-based Combination Therapy in Papua, Indonesia' has been provisionally accepted for publication in PLOS Neglected Tropical Diseases.

Best regards,

Gregory Deye

Guest Editor

Walderez Dutra

Deputy Editor

---

## [Editor Report · Acceptance letter]

28 Apr 2020

Dear Dr Auburn,

We are delighted to inform you that your manuscript, "Molecular surveillance over 14 years confirms reduction of Plasmodium vivax and falciparum transmission after implementation of Artemisinin-based Combination Therapy in Papua, Indonesia," has been formally accepted for publication in PLOS Neglected Tropical Diseases.

Best regards,

Serap Aksoy

Editor-in-Chief

Shaden Kamhawi

Editor-in-Chief
